Association between lactic acidosis and multiple organ dysfunction syndrome after cardiopulmonary bypass

Zheng Dan 1
Yu Guo-Liang 1
Zhou Yi-Ping 1
Zhang Qiao-Min 1
Wang Chun-Guo 2
Zhang Sheng 13676655525@163.com 1
1 Department of Critical Care Medicine, Taizhou Hospital of Zhejiang Province, Wenzhou Medical University , Linhai , China
2 Department of Cardiothoracic Surgery, Taizhou Hospital of Zhejiang Province, Wenzhou Medical University , Linhai , China
Palamaner Subash Shantha Ghanshyam
Electronic publication date: 2024 Jan 31
Publication date: 2024
Volume: 12
Electronic Location ID: e16769
Received 2023 Jun 16; Accepted 2023 Dec 16
Copyright: ©2024 Zheng et al.
Copyright year: 2024
Copyright holder: Zheng et al.
License: This is an open access article distributed under the terms of the Creative Commons Attribution License, which permits unrestricted use, distribution, reproduction and adaptation in any medium and for any purpose provided that it is properly attributed. For attribution, the original author(s), title, publication source (PeerJ) and either DOI or URL of the article must be cited.
License URL: https://creativecommons.org/licenses/by/4.0/

Keywords: Cardiopulmonary bypass, Acidosis, Lactic, Hyperlactatemia, Multiple organ dysfunction syndrome

Funding: The Medical and Health Science and Technology Project of Zhejiang Province 2017KY163 This study was supported by the Medical and Health Science and Technology Project of Zhejiang Province (2017KY163). The funders had no role in study design, data collection and analysis, decision to publish, or preparation of the manuscript.

==============================
Background

The relationship between hyperlactatemia and prognosis after cardiopulmonary bypass (CPB) is controversial, and some studies ignore the presence of lactic acidosis in patients with severe hyperlactacemia. This study explored the association between lactic acidosis (LA) and the occurrence of multiple organ dysfunction syndrome (MODS) after cardiopulmonary bypass.

Methods

This study was a post hoc analysis of patients who underwent cardiac surgery between February 2017 and August 2018 and participated in a prospective study at Taizhou Hospital. The data were collected at: ICU admission (H0), and 4, 8, 12, 24, and 48 h after admission. Blood lactate levels gradually increased after CPB, peaking at H8 and then gradually decreasing. The patients were grouped as LA, hyperlactatemia (HL), and normal control (NC) based on blood test results 8 h after ICU admission. Basic preoperative, perioperative, and postoperative conditions were compared between the three groups, as well as postoperative perfusion and oxygen metabolism indexes.

Results

There were 22 (19%), 73 (64%), and 19 (17%) patients in the LA, HL, and NC groups, respectively. APACHE II (24h) and SOFA (24h) scores were the highest in the LA group (P < 0.05). ICU stay duration was the longest for the LA group (48.5 (42.5, 50) h), compared with the HL (27 (22, 48) h) and NC (27 (25, 46) h) groups (P = 0.012). The LA group had the highest incidence of MODS (36%), compared with the HL (14%) and NC (5%) groups (P = 0.015). In the LA group, the oxygen extraction ratio (O2ER) was lower (21.5 (17.05, 32.8)%) than in the HL (31.3 (24.8, 37.6)%) and the NC group (31.3 (29.0, 35.4) %) (P = 0.018). In the univariable analyses, patient age (OR = 1.054, 95% CI [1.003–1.109], P = 0.038), the LA group (vs. the NC group, (OR = 10.286, 95% CI [1.148–92.185], P = 0.037), and ΔPCO2 at H8 (OR = 1.197, 95% CI [1.022–1.401], P = 0.025) were risk factor of MODS after CPB.

Conclusions

We speculated that there was correlation between lactic acidosis and MODS after CPB. In addition, LA should be monitored intensively after CPB.

Introduction

Cardiac surgery comes with a significant risk of major adverse events, with an overall mortality rate of 2%–3% (Senst, Kumar & Diaz, 2022). Major complications include bleeding, stroke, kidney injury, mesenteric ischemia, atrial fibrillation, cardiogenic shock, and respiratory distress (Senst, Kumar & Diaz, 2022). Multiple organ dysfunction syndrome (MODS) occurs in about 5% of patients after cardiac surgery, with cardiopulmonary bypass (CPB) being the procedure with the highest risk (D’Agostino et al., 2018; Liu et al., 2021). The 28-day mortality rate of MODS is 23% (Stoppe et al., 2016), and MODS can persist for as long as 28 days in 15% of patients (Stoppe et al., 2016). MODS after cardiac surgery is associated with high morbidity and mortality. Predicting the risk of MODS after cardiac surgery could help screen patients requiring closer management.

Hyperlactatemia occurs in 10%–20% of patients following cardiac surgeries (Minton & Sidebotham, 2017), as it can result from hypoxic and nonhypoxic causes such as drug therapy, cardioplegia, hypothermia, and CPB. Severe hyperlactatemia can lead to lactic acidosis (LA; Garcia-Camacho et al., 2020; Greenwood et al., 2021; Teloh et al., 2017), which can then lead to vascular hyporeactivity, decreased peripheral vascular resistance, decreased myocardial contraction force, and reactivation of the heart and exogenous catecholamines, causing myocardial injury, severe arrhythmia, inflammation, and impaired immune cell function (Azad, Islam & Quasem, 2019; Kimmoun et al., 2015; Rudnick et al., 2020). The relationship between hyperlactatemia and prognosis after CPB remains controversial, however, and the limits of lactic acid values for hyperlactacemia are not uniform in published research (Joudi et al., 2014; Mak et al., 2016; Minton & Sidebotham, 2017). Normal lactate is generally defined as blood lactate <2.0 mmol/L (Foucher & Tubben, 2022), but because there is no standard definition for hyperlactatemia, it has been variably defined as blood lactate >3 or >4 mmol/L, or even >10 mmol/L (Bhardwaj et al., 2017; Kedziora et al., 2020; Renew et al., 2016). LA is a well-known marker of poor tissue perfusion and multiorgan dysfunction, and LA can predict mortality in patients with severe sepsis and septic shock and the incidence of MODS in hospitalized patients (Doshi et al., 2018; Gattinoni et al., 2019). LA and a longer time to lactic acid normalization are also associated with an increased risk of death (Bullock & Benham, 2022; Gillies et al., 2019; Radelfahr & Klopstock, 2019). However, there are few studies on the effect of LA on MODS after CPB. This study aimed to determine the association between LA and the occurrence of MODS after CPB. The results of this study could help improve the management of CPB patients.

Materials & Methods

Study design and participants

This study was a post hoc analysis of the clinical data from all patients who underwent cardiac surgery between February 2017 and August 2018 and participated in a prospective study (monitoring oxygen metabolism after cardiopulmonary bypass and 48-h organ dysfunction, ChiCTR-ROC-1701072; Zhang et al., 2021). This study was approved by the ethics committee of Taizhou Hospital (Zhejiang Province, China) and was carried out according to the tenets of the Helsinki Declaration of 1975 and good clinical practice. Written informed consent was obtained from all patients in the original study (Zhang et al., 2021), including consent to the use of study data for post hoc analyses.

The study inclusion criteria were: (1) ≥18 years of age, (2) admission to the ICU immediately after CPB, and (3) ICU stay of more than one day. The exclusion criteria were: (1) received NaHCO3−, (2) emergency surgery, (3) pregnancy, (4) chronic renal insufficiency with dialysis, (5) preoperative acute or chronic liver failure, (6) hematologic diseases, (7) central venous catheter dislocation, or (8) death within 48 h. Intraoperative and postoperative ICU management was conducted in accordance with local protocols and international guidelines (Carl et al., 2010; Zhang et al., 2021).

Data collection and definitions

Intraoperative and ICU management was performed using the same methods as the original study (Zhang et al., 2021). The following data were collected: basic patient characteristics before surgery, intraoperative conditions, postoperative vital signs, and blood test results. The data were collected at: ICU admission (H0), and 4 (H4), 8 (H8), 12 (H12), 24 (H24), and 48 h (H48) after admission. The Acute Physiology and Chronic Health Evaluation II (APACHE II) and sequential organ failure (SOFA) scores were assessed between 24 and 48 h after ICU admission. A blood gas analysis of arteries and veins were performed at ICU admission, and at H4, H8, H12, and H24 using an automated analyzer (ABL800 Flex, Radiometer Medical Aps, Bronshoj, Denmark).

Blood lactate levels gradually increased after CPB, peaking at H8 and then gradually decreasing (Fig. 1). In animal studies, lactic acid level increased after CPB, peaking at 8–12 h and gradually decreasing to the normal range 24 h after surgery (Chiang et al., 2001). A previous clinical study found that lactic acid level peaked 6–9 h after surgery and then gradually decreased, with most levels returning to normal within 24 h (Aktar et al., 2020). Based on these results, H8 was selected for grouping. Patients were divided into three groups according to blood lactate level, HCO3−, and pH measurement at H8: the lactic acidosis group (LA group; blood lactate levels ≥ 5 mmol/L, pH < 7.35, and HCO3− < 20 mmol/L), the hyperlactatemia group (HL group; blood lactate levels > 2 mmol/L, without pH < 7.35 and HCO3− < 20 mmol/L), and the normal control group (NC group; blood lactate levels ≤ 2 mmol/L, without pH < 7.35 and HCO3− < 20 mmol/L (Kraut & Madias, 2014).

Figure 1 Blood lactate level trends during the 24-h observation period.

H8 lactate (Lac), H12 Lac, and H0 Lac were statistically significant (P = 0.001, 0.007).

Outcomes

The primary outcome was the occurrence of MODS at H48. The diagnostic indexes of organ dysfunction used in this study refer to the same criteria as the original study, and MODS was defined as the dysfunction of two or more organs (Zhang et al., 2021). Organ dysfunction within 48 h after surgery was diagnosed in accordance with the following criteria: the diagnostic criteria for acute respiratory distress syndrome (ARDS) was a PaO2/FIO2 of <300 mmHg requiring non-invasive ventilation or invasive mechanical ventilation support; acute kidney injury was defined according to Kidney Disease Improving Global Outcomes (KDIGO): Clinical Practice Guideline for Acute Kidney Injury; additional criteria included total bilirubin levels ≥32 µmmol/L, platelet count <100 × 109/L, and acute tissue hypoperfusion (presence of tachycardia and hypotension, with central venous oxygen saturation (ScvO2) level of <65% and cardiac index of ≤2.2 L/min/m2), cardiac arrest, arrhythmia (ventricular tachycardia and ventricular fibrillation), and acute neurological dysfunction (stroke, seizure, persistent delirium, and Glasgow coma score below 12).

Statistical analysis

The statistical analysis was conducted using SPSS 2.0 (IBM Corp., Armonk, NY, USA). The Kolmogorov–Smirnov test was used to evaluate the normality of the continuous variables and the results showed that many continuous variables were non-normally distributed. All continuous variables were expressed as median (range) and analyzed using the Mann–Whitney U-test, the Kruskal–Wallis test (intergroup comparisons), or repeated-measures ANOVA (intragroup comparisons). Categorical variables were presented as m (%) and analyzed using the chi-square test. The factors associated with MODS at H48 were analyzed using a multivariable logistic regression analysis. Variables with P < 0.10 in the univariable regression analyses were included in the multivariable regression analysis. GraphPad Prism version 8.0.2 (GraphPad Software Inc., San Diego, CA, USA) was used for the boxplot charts.

Results

Clinical characteristics and prognosis of the patients

The original study enrolled 139 patients, but 25 were excluded based on the exclusion criteria of the present study, and eight patients were excluded due to incomplete data. Therefore, 114 patients were included in this study: 22 (19%) in the LA group, 73 (64%) in the HL group, and 19 (17%) in the NC group (Fig. 2). There were no significant differences in the preoperative and intraoperative variables among the three groups (all P > 0.05). The APACHE II (H24) and SOFA (H24) scores were the highest in the LA group (P < 0.05). The ICU stay duration was the longest for the LA group, at 48.5 h (42.5 h, 50 h), compared with 27 h (22 h, 48 h) for the HL group and 27 h (25 h, 46 h) for the NC group (P = 0.012). The patients in the LA group had the highest incidence of MODS (36%), compared with 14% in the HL group and 5% in the NC group (P = 0.015; P < 0.05 for LA vs. HL and LA vs. N; Table 1).

Vital signs and perfusion variable at H8

The pH, HCO3−, and lactate levels were all significantly different between the three groups (all P < 0.001). The LA group had lower BE (−7.6 (−9.25, −6.05) vs. −2.3 (−4.4, −1.05) vs. −1.35 (−2.5, 0.75)), lower SaO2-SvO2 (21.0 (16.5, 32.5) vs. 31.0 (23.75, 37.0) vs. 31.0 (29.0, 36.5)) and lower O2ER (21.5 (17.05, 32.8) vs. 31.3 (24.8, 37.6) vs. 31.3 (29.0, 35.4)) compared with the HL and NC groups (all P < 0.05). The LA group had higher ScvO2 compared with the NC group (76 (66.5, 81) vs. 68 (64, 70), P = 0.024) (Table 2).

Table 1 Patients’ clinical characteristics and prognosis.

Subject	LA group (n = 22; H8)	HL group (n = 73; H8)	NC group (n = 19; H8)	P	
Preoperative					
Age (years)	59.50 (55.75, 68.25)	57.00 (48.50, 64.00)	60.00 (50.00, 66.00)	0.421	
Sex (male/female)	6/16	34/39	12/7	0.068	
Body mass index (kg/m2 )	22.59 (19.56, 24.57)	22.77 (20.83, 25.38)	23.07 (20.85, 24.23)	0.633	
Smoking (yes/no)	3/19b	26/47	10/9	0.029	
Diabetes (yes/no)	0/22	6/67	2/17	0.336	
Hypertension (yes/no)	8/14	27/46	10/9	0.437	
Tumor (yes/no)	1/21	3/70	0/19	0.658	
COPD (yes/no)	1/21	1/72	1/18	0.527	
Coronary heart disease (yes/no)	4/18	4/69	0/19	0.052	
Cerebrovascular disease (yes/no)	2/20	3/70	1/18	0.657	
EF (%)	59.5(54,64.25)	63(57,67)	64(53,67)	0.196	
pH	7.41 (7.39, 7.42)	7.40 (7.38, 7.42)	7.40 (7.39, 7.41)	0.551	
BE (mmol/L)	−0.5 (−1.6, 1.3)	−1.0 (−1.9, 0.5)	1.0 (−0.9, 1.8)	0.351	
Hb (g/L)	136 (126, 148)	135 (126, 149)	139 (128, 149)	0.743	
PLT count (109 /L)	222 (165, 264)	211 (171, 245)	204 (161, 254)	0.385	
ALT (U/L)	25 (14, 36)	20 (12, 40)	25 (16, 35)	0.796	
AST (U/L)	22 (20, 27)	22 (20, 31)	29 (24, 33)	0.054	
TBIL (µmol/L)	15 (10.5, 16.6)	12.2 (8.6, 17.4)	9.7 (7.5, 16.2)	0.287	
DBIL (µmol/L)	4.6 (3.8, 5.6)	4.3 (2.9, 5.8)	3.3 (2.4, 6.0)	0.486	
Albumin (g/L)	40 (37.5, 44.6)	42.9 (39.9, 46.3)	43 (39.3, 44.4)	0.165	
SCr (µmol/L)	69 (61, 87)	79 (72, 85)	74.5 (61.5, 84)	0.351	
BUN (mmol/L)	5.9 (4.6, 8.7)	6.2 (4.8, 7.7)	7.5 (5.8, 8.3)	0.538	
GLU (mmol/L)					
Surgery					
Valve replacement	14	56	13	0.43	
Others	8	17	6		
Duration of surgery (h)	4.5 (3.5, 5.0)	4.0 (3.5, 5.0)	3.6 (3.1, 4.2)	0.354	
CBP duration (min)	110 (60, 135)	105 (85, 140)	110 (75, 134)	0.911	
Aortic clamping duration (min)	60 (42, 75)	65 (50, 85)	55 (30, 82.5)	0.829	
Prognosis					
APACHEII (H24) score	10 (9, 13)a,b	7 (4,9)	7 (5, 10)	0.003	
SOFA (H24) score	7 (5, 8) a,b	5 (3, 7)	5 (4, 6)	0.005	
Ventilation duration (h)	16.5 (13.5, 17.5)	16.6 (13.5, 17.5)	17 (13.5, 18)	0.369	
ICU stay duration (h)	48.5 (42.5, 50)a	27 (22, 48)	27 (25, 46)	0.012	
Hospital stay duration (d)	11 (8, 13)	10 (9, 13)	11.5 (8.5, 17)	0.456	
Organ dysfunction (yes/no)	11/11	31/42	6/13	0.489	
Multiple organ dysfunction (yes/no)	8/14a,b	10/63	1/18	0.015	
Notes.

APACHE II Acute Physiology and Chronic Health Evaluation II

ALT alanine aminotransferase

AST aspartate aminotransferase

BMI body mass index

BE base excess

BUN Urea nitrogen

CPB cardiopulmonary bypass

DBIL direct bilirubin

H24 24 h after entering the ICU

H48 48 h after entering the ICU

HB hemoglobin

PLT platelet

SCr sSerum creatinine

SOFA Sequential Organ Failure Assessment

TBIL total bilirubin

a P < 0.05 between the LA and HL groups.

b P < 0.05 between the LA and NC groups.

c P < 0.05 between the HL and NC groups.

Table 2 Vital signs and perfusion variable at H8.

	LA group (n = 22; T8)	HL group (n = 73; T8)	NC group (n = 19; T8)	P	
SBP (mmHg)	100.00 (92.5, 112.5)	106.00 (100.00, 112.25)	98.5 (93.5, 111.25)	0.106	
DBP (mmHg)	55.0 (49.5, 60.5)	59.0 (52.75, 64.0)	57.5 (51.25, 63.25)	0.169	
MAP (mmHg)	72.0 (63.5, 79.0)	74.0 (68.75, 79.0)	69.5 (67.00, 75.75)	0.196	
CVP (mmHg)	10 (7.5, 12.0)	7.0 (6.0, 9.25)	9.0 (7.5, 10.0)	0.033	
VIS	4.00 (0.00, 8.00)	3.0 (0.00, 9.00)	8.0 (2.00, 9.00)	0.377	
Intake (mL)	1184 (1057.5, 1592.5)	1085 (861.75, 1445)	1150 (860.0, 1551.25)	0.394	
Output (mL)	871 (721.5, 1266)b	952.5 (843.75, 1226.25)	1242 (768.75, 1707.5)	0.028	
Urine volume (mL)	655 (544.0, 1033.5)b	782.5 (550.00, 973.75)	1062.5 (653.5, 1367.5)	0.012	
Fluid balance (mL)	340.0 (−153.75, 623.5)	33.0 (−187.0, 367.5)	−20 (−395, 185)	0.093	
pH	7.29 (7.23, 7.31)a,b	7.39 (7.36, 7.43)	7.39 (7.37, 7.44)	<0.001	
BE (mmol/L)	−7.6 (−9.25, −6.05)a,b	−2.3 (−4.4, −1.05)	−1.35 (−2.5, 0.75)	0.001	
HCO3 (mmol/L)	19.2 (18.5, 20.5)a,b	22.0 (20.3, 23.6)c	24.1 (22.4, 25.6)	<0.001	
Lactate (mmol/L)	6.8 (5.6, 9.6)a,b	4.1 (3.0, 5.4)c	1.7 (1.3, 1.9)	<0.001	
ScvO2 (%)	76 (66.5, 81)b	68 (61.0, 74.5)	68 (64, 70)	0.024	
SaO2-SvO2 (%)	21.0 (16.5, 32.5)a,b	31.0 (23.75, 37.0)	31.0 (29.0, 36.5)	0.016	
O2ER (%)	21.5 (17.05, 32.8)a,b	31.3 (24.8, 37.6)	31.3 (29.0, 35.4)	0.018	
△PCO2 (mm Hg)	10.00 (6.50, 11.50)	9.50 (7.75, 11.00)	9.00 (5.00, 11.00)	0.417	
△PCO2/Ca-vO2 (mm Hg/mL)	2.1 (1.8, 3.7)	2.0 (1.6, 2.3)	1.8 (1.3, 2.1)	0.084	
Notes.

BE base excess

CVP central venous pressure

Ca-vO2 arterial-to-central venous oxygen content difference

DBP diastolic blood pressure

MAP mean arterial pressure

O2ER oxygen extraction ratio

△PCO2 central venous-to-arterial carbon dioxide difference

△PCO2/C(a-cv)O2 △PCO2-to-arterial-to-central venous oxygen content difference rate

SBP systolic blood pressure

ScvO2 central venous oxygen saturation

SaO2-SvO2 arterial-to-central venous oxygen saturation difference

VIS vasoactive-inotropic score

a P < 0.05 between the LA and HL groups.

b P < 0.05 between the LA and NC groups.

c P < 0.05 between the HL and NC groups.

Risk factors associated with MODS

In the univariable analyses, patient age (OR = 1.054, 95% CI [1.003–1.109], P = 0.038), the LA group (vs. NC, OR = 10.286, 95% CI [1.148–92.185], P = 0.037), and ΔPCO2 at H8 (OR = 1.197, 95% CI [1.022–1.401], P = 0.025) were associated with MODS after CPB (Table 3).

Table 3 The risk factors of MODS were analyzed, and the perfusion index at the H8 time point was analyzed by binary logistic regression.

Subject	Univariate	
	OR (95% CI)	P	
Group			
NC	Reference		
HL vs. NC	2.857 (0.342–23.835)	0.332	
LA vs. NC 	10.286 (1.148–92.185)	0.037	
Age	1.054 (1.003–1.109)	0.038	
ScvO2 (H8)	0.953 (0.908–1.000)	0.051	
O2ER (H8)	1.023 (0.995–1.052)	0.108	
△PCO2 (H8)	1.197 (1.022–1.401)	0.025	
△PCO2/Ca-vO2 (H8)	1.218 (0.849–1.746)	0.283	
Notes.

P < 0.05 was considered statistically significant.

MODS multiple organ dysfunction syndrome

OR odds ratio

CI confidence interval

NC normal control

HL hyperlactatemia

LA lactic acidosis

ScvO2 central venous oxygen saturation

SaO2-SvO2 arterial-to-central venous oxygen saturation difference

O2ER oxygen extraction ratio

△PCO2 central venous-to-arterial carbon dioxide difference

△PCO2/C(a-cv)O2 △PCO2-to-arterial-to-central venous oxygen content difference rate

Figure 2 Flowchart.

Chronic kidney disease (CKD).

Discussion

This post hoc analysis of a previous prospective study aimed to determine the association between LA and the occurrence of MODS after CPB. The LA group had long ICU stays and a high incidence of MODS, suggesting that LA was associated with MODS after CPB.

Tissue hypoperfusion and lactic acid accumulation can occur during cardiac surgery (Chiang et al., 2001; Kanazawa et al., 2015; Naik et al., 2016; Teloh et al., 2017). When tissue perfusion is restored after CPB, lactic acid displays a washout effect, and a large amount of lactic acid is released into the blood, leading to apparent hyperlactatemia, but that lactic acid is cleared rapidly if the heart and vital organs recover well after surgery, preventing LA (Kanazawa et al., 2015). LA occurs when the body fails to compensate (Kanazawa et al., 2015). Some studies suggest that high lactacidemia during operation is related to CPB time and aortic occlusion time (Naik et al., 2016; Ranucci et al., 2006). In this study, lactic acid levels were grouped eight hours after surgery, and there was no difference in cardiopulmonary bypass time and aortic occlusion time among the three groups. This may be because the patients were stable before surgery and the operations were performed by the same surgeon, so there was no difference in time of CPB and aortic occlusion. Naik et al. (2016) reported that the incidence of hyperlactatemia (blood lactate levels ≥ 4 mmol/L) was 42.7%, and these patients had an increased incidence of postoperative atrial fibrillation and extended ICU stays. Renew et al. (2016) observed that severe hyperlactatemia (blood lactate level > 10 mmol/L) was associated with increased mortality. Zante et al. (2018) reported that blood lactate levels >4.0 mmol/L, combined with a BE <6.7 mmol/L, could predict mortality, whereas blood lactate levels >4.0 mmol/L alone did not affect mortality after cardiac surgery. However, these studies ignored the presence of lactic acidosis in patients with severe hyperlactacemia, especially when the blood lactate levels were >5 mmol/L. In the present study, the blood lactate levels were 6.8 (5.6, 9.6) mmol/L in the LA group and 4.1 (3.0, 5.4) mmol/L in the HL group. The LA group tended to be severely ill, had a prolonged ICU stay, and a high incidence of MODS. Hyperlactemia alone did not affect prognosis.

Clinically, ScvO2, O2ER, ΔPCO2, and ΔPCO2/Ca-vO2 are often used to evaluate tissue perfusion (Janotka & Ostadal, 2021). The O2ER is the amount of oxygen taken out of the blood by the tissues. Normal O2ER is about 25%–30%. In this study, O2ER was calculated as 1-ScvO2/SaO2. In sepsis, because oxygen free radicals and inflammatory mediators damage the structure of the mitochondria, the number and function of mitochondria declines. The longer severe sepsis lasts, the more irreversible mitochondrial structure and function changes occur, resulting in mitochondrial dysfunction and microcirculation disorders, decreased oxygen use, decreased oxygen consumption, increased ScvO2, and reduced O2ER (Gong & Li, 2013; Pope et al., 2010). In a study of patients with septic shock, the hospital mortality rate was 41.7% in the low O2ER group, which was higher than the normal and high O2ER groups. O2ER was negatively correlated with ScvO2 (R2 = 0.878, P < 0.001), and high ScvO2 combined with low O2ER was associated with mitochondrial dysfunction (Park et al., 2015). In the present study, there was no difference in mean arterial pressure between the three groups, but the LA group had decreased urine volume, increased ScvO2, and decreased O2ER, due to the SIRS produced by CPB, and the oxygen free radicals and various chemia-reperfusion injuries leading to mitochondrial dysfunction, causing lactic acidosis (Balzer et al., 2015; Janotka & Ostadal, 2021; Stephens et al., 2020). Dekker et al. (2019) found persistent microcirculation perfusion disorders through hypoglossal flow dark field microcirculation imaging technology in the first three days after CPB. A separate study found that microcirculation disorders persisted 24 h after CPB through the reduction of functional capillary density (FCD), perfusion vessel density (PVD), or perfusion vessel proportion (PPV), even when general circulation was stable (den Os et al., 2020). A single-center study analyzed the relationship between microcirculation heterogeneity and hemodynamics and found that SaO2-SvO2 and O2ER decreased in microcirculation perfusion disorders after CPB (Koning et al., 2014). In the present study, there were high ScvO2 and low O2ER in the LA group, suggesting that LA after CPB may be accompanied by microcirculation and mitochondrial dysfunction. MODS can be the result of both hypoperfusion and hypo-oxygenation (Rixen & Siegel, 2005), or the result of microcirculation and mitochondrial dysfunction, reflected by LA (Garcia-Camacho et al., 2020; Greenwood et al., 2021; Teloh et al., 2017). Even if the patient’s circulation is stable after CPB, there might be mitochondrial dysfunction and microcirculation disorders, resulting in organ dysfunction. However, this study was a retrospective analysis, and the indicators of mitochondrial function could not be analyzed with LA. Therefore, future studies are needed to evaluate this relationship.

In this study, patient age and △PCO2 were risk factors for MODS, but the OR value of age and △PCO2 were close to 1, and were of little clinical significance. The research team previously found that ΔPCO2 and ΔPCO2/Ca-vO2 levels are not reliable predictors of organ dysfunction occurrence at H48 (Lefering et al., 2013; Zhang et al., 2021). LA is a prognostic factor for MODS in sepsis (Bakker et al., 1996; Doshi et al., 2018; Gattinoni et al., 2019), trauma with massive hemorrhage (Lefering et al., 2013), and cardiogenic shock (Jentzer et al., 2022). The occurrence of MODS leads to longer ICU stays, higher hospital expenses, and increased morbidity and mortality (Asim, Amin & El-Menyar, 2020; Zhao et al., 2022). In this study, LA was associated with MODS. However, due to the insufficient number of cases in this study, no further multi-factor analysis was done. Because LA can be detected quickly through a blood gas analysis, LA could be used as a monitoring indicator of MODS after CPB. There are many causes of lactic acidosis, including tissue hypoperfusion, hyperglycemia, and endogenous catecholamines caused by the stress response during cardiac surgery. The liver accounts for 60% of lactic acid metabolism, and the kidney accounts for 30% (Lefering et al., 2013; Stephens et al., 2020). Lactic acidosis could be avoided by controlling blood glucose and body temperature, shortening the time of aortic blood flow blockade during operation, optimizing volume management, supporting organ function treatment, and ensuring the perfusion of macrocirculation and microcirculation after operation. LA could be used as a reference index for the occurrence of MODS after CPB, better guiding clinical treatment for post-surgical patients.

Limitations

This study has some limitations. First, this was a post hoc analysis of a completed prospective study. The sample size of the original study was limited, and some patients had to be excluded from the present analysis. This small sample size likely led to the wide 95% CI observed in the association between LA and MODS in the univariate analysis. Therefore, no further multivariate regression analysis was performed. Second, it was a single-center study, and generalizability might be limited. Third, this study did not include data on whether LA after cardiac surgery was only caused by increasing lactic acid levels or by other factors. Therefore, a future prospective multicenter study is planned, with a larger sample size, to collect relevant data with MODS as the outcome indicator, monitoring mitochondrial function indexes and investigating the relationship between LA and MODS.

Conclusion

This study identified a correlation between lactic acidosis and MODS after CPB. LA is an easy-to-obtain clinical indicator, so LA should be closely monitored after CPB to help improve patient prognosis after CPB.

Supplemental Information

Supplemental Information 1 Raw data

Click here for additional data file.

The authors kindly thank Dr. Zhang Meixian for her contribution to establishing the database.

Additional Information and Declarations

Competing Interests

Author Contributions

Human Ethics

Data Availability

The authors declare there are no competing interests.

Dan Zheng conceived and designed the experiments, authored or reviewed drafts of the article, and approved the final draft.

Guo-Liang Yu conceived and designed the experiments, analyzed the data, authored or reviewed drafts of the article, and approved the final draft.

Yi-Ping Zhou performed the experiments, prepared figures and/or tables, and approved the final draft.

Qiao-Min Zhang performed the experiments, prepared figures and/or tables, and approved the final draft.

Chun-Guo Wang analyzed the data, prepared figures and/or tables, and approved the final draft.

Sheng Zhang conceived and designed the experiments, authored or reviewed drafts of the article, and approved the final draft.

The following information was supplied relating to ethical approvals (i.e., approving body and any reference numbers):

The study was approved by the ethics committee of Taizhou Hospital (Zhejiang Province, China) (Ethical Application No. ChiCTR-ROC-17010727)

The following information was supplied regarding data availability:

The raw data are available in the Supplementary File.

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
