# Peer review of "Association between lactic acidosis and multiple organ dysfunction syndrome after cardiopulmonary bypass"

_PeerJ, doi:10.7717/peerj.16769_

## Round 0.1 · original submission · Major Revisions

Thank you for the opportunity to review your manuscript. I agree with the reviewers except for the comment on the assessment of malignancy in this patient subset. I do not believe malignancy assessment is important to this population.

I recommend the authors to give a good effort at making sure grammar errors are avoided. Also, I do agree that the multivariate logistic regression is adequately powered for the small sample size. Hence, recommend omitting the multivariate analysis. Also, I am unsure of the novelty of the paper. It is well known that lactic acidosis and accompanying hemodynamic instability predict poor outcomes in any situation. I am unsure how this study adds to the literature. I request the authors to discuss the novelty of the paper and its contribution to the literature.

Reviewer 1 ·

Basic reporting

Referencing and journal guideline is not followed in some instances.

In-text citations
For three or fewer authors, list all author names (e.g. Smith, Jones & Johnson, 2004). For four or more, abbreviate with ‘first author’ et al. (e.g. Smith et al., 2005).

Experimental design

I read “Association between lactic acidosis and multiple organ dysfunction syndrome after cardiopulmonary bypass” by Dr. Zheng et al. with great curiosity and enthusiasm.
I commend the authors for their extensive data compiled from their institutional research. In addition, the manuscript is clear; however, some concern needs to be addressed before proceeding.
Major issues
There are essential issues in “Sample size considerations.” Calculating the sample size required for running logistic regression is a crucial element. Peduzzi et al. (1996) proposed guidelines for calculating the minimum number of cases to include in the analysis. Let p be the smallest of the proportions of negative or positive patients in the population and k the number of covariates (the number of independent variables). The minimum number of cases to include is N = 10 k / p. The current study did not have a sufficient sample size to perform a logistic regression model. They have six covariates included in the multivariate model, and the proportion of deaths cases is 0.2. therefore, the minimum number of cases required is at least 300, not 114 cases!!!!!!!
Due to the above issues, the CI is unusually wide, so the final result with an appropriate sample size may go in any direction.
Based on the above limitations with other shortcomings mentioned by the authors in limitations (single-center, non-randomized study with chances of confounding bias due to other factors), I’m afraid I have to disagree with the conclusion with a hard statement mentioning “LA was an independent risk factor of MODS after CPB”, which may or may not hold true with an increase in sample size.

Minor issues
Authors may add more methodical details in the methods of the abstract so they would be clearer to readers/reviewers.
Authors could rather state a more robust/ possibly controlled study to test the association of LA and MODS hypothesis based on their findings and state as future direction. This can open the door for more studies to test the hypothesis with proper sample calculation and methodical consideration for the research question.

Validity of the findings

I read “Association between lactic acidosis and multiple organ dysfunction syndrome after cardiopulmonary bypass” by Dr. Zheng et al. with great curiosity and enthusiasm.
I commend the authors for their extensive data compiled from their institutional research. In addition, the manuscript is clear; however, some concern needs to be addressed before proceeding.
Major issues
There are essential issues in “Sample size considerations.” Calculating the sample size required for running logistic regression is a crucial element. Peduzzi et al. (1996) proposed guidelines for calculating the minimum number of cases to include in the analysis. Let p be the smallest of the proportions of negative or positive patients in the population and k the number of covariates (the number of independent variables). The minimum number of cases to include is N = 10 k / p. The current study did not have a sufficient sample size to perform a logistic regression model. They have six covariates included in the multivariate model, and the proportion of deaths cases is 0.2. therefore, the minimum number of cases required is at least 300, not 114 cases!!!!!!!
Due to the above issues, the CI is unusually wide, so the final result with an appropriate sample size may go in any direction.
Based on the above limitations with other shortcomings mentioned by the authors in limitations (single-center, non-randomized study with chances of confounding bias due to other factors), I’m afraid I have to disagree with the conclusion with a hard statement mentioning “LA was an independent risk factor of MODS after CPB”, which may or may not hold true with an increase in sample size.

Minor issues
Authors may add more methodical details in the methods of the abstract so they would be clearer to readers/reviewers.
Authors could rather state a more robust/ possibly controlled study to test the association of LA and MODS hypothesis based on their findings and state as future direction. This can open the door for more studies to test the hypothesis with proper sample calculation and methodical consideration for the research question.

Additional comments

as above

Reviewer 2 ·

Basic reporting

There were a number of grammar errors through the paper. Abbreviation should also be consistent through the paper. Please have someone that is proficient in English review the paper and correct grammar errors. However, I commend authors for presenting the study in a professional manner.

Experimental design

The concept of LA after CPB is interesting though there is extensive literature on lactic acidosis in ICU patients.

I think there should be a clear explanation on collection, processing, transport to lab, time frame lactic acid assay was run as variation in the process can result in erroneous LA levels and impact the validity of the study.

Validity of the findings

I found baseline comparison of clinical characteristics anemic. In order to derive something meaningful from the different cohorts, a more robust baseline characteristic comparison is necessary. For example, include important characteristics that affect blood flood and consequently LA levels such as PAD and history of CVA to name a couple. Ejection fraction prior to surgery is also important to compare between groups. Smoking status should also be included. In my opinion, comparison of the type of CPB surgery is also a vitally important baseline characteristic.

Will the addition of these variables and more, I think the likelihood of type 1 error is less likely.

As the study sits now, I don't think we can say the groups are even with any degree of certainty which makes the conclusion of the study much less meaningful.

Additional comments

The ethics approval document should be in English so I can verify the document.

Reviewer 3 ·

Basic reporting

fairly well written

Experimental design

Small sample size.

Validity of the findings

No comment - it may need to be reviewed by a statistician.

Additional comments

I have reviewed the study -- Association between lactic acidosis and multiorgan dysfunction after cardiopulmonary bypass -- it is a post hoc prospective study of 114 patients which showed lactic acidosis after Cardiopulmonary bypass as an independent predictor of multiorgan dysfunction.
My comments -
1. Unfortunately, the study has a small sample size of 114 patients and out of which it includes only 22 patients with lactic acidosis, leading to bias
2. There are multiple causes of lactic acidosis, including Malignancy - were patients with active malignancy are excluded ?.
3. Were any patients on Metformin before the surgery? Which could also be a cause of lactic acidosis/ elevated lactate
4. Longer duration of surgery and lactic acidosis compared to normal subjects? Can the author comment on the finding
5. It is well known lactic acidosis is a marker of poor tissue perfusion and multiorgan dysfunction -- can the author comment on what strategies pre and perioperatively can be done to prevent lactic acidosis

---

## Round 0.2 · accepted · Accept

I thank the authors for their efforts in revising this manuscript. The paper reads much better. There are limitations inherent to the study design and sample size. But I agree that the paper will add to existing literature and hence deserves publication.

Reviewer 1 ·

Basic reporting

Manuscript has been revised substantially addressing prior flaws, can be accepted and send for production with minor layout edits before final publication

Experimental design

Manuscript has been revised substantially addressing prior flaws, can be accepted and send for production with minor layout edits before final publication

Validity of the findings

Manuscript has been revised substantially addressing prior flaws, can be accepted and send for production with minor layout edits before final publication

Additional comments

None

Reviewer 3 ·

Basic reporting

corrected grammar and spelling mistakes after revision

Experimental design

no comments

Validity of the findings

changes made in the revised study -removed multivariate analysis
Would definitely recommend statistical analysis by expert

Additional comments

I have reviewed the revised manuscript -- Association between lactic acidosis and multiorgan dysfunction after cardiopulmonary bypass -- it is a post hoc prospective study that showed lactic acidosis after Cardiopulmonary bypass as an independent predictor of multiorgan dysfunction.
My comments
The authors have provided excellent responses to questions by reviewers and made necessary changes.
Lactic acidosis is a well-known poor prognostic marker --further. This small sample size study validates it.
We need a large sample multicenter study with early intervention based on lactic acid levels to improve prognosis.